# Effects of a Person Centered Dementia Training Program in Greek Hospital Staff—Implementation and Evaluation

**DOI:** 10.3390/brainsci10120976

**Published:** 2020-12-12

**Authors:** Mara Gkioka, Birgit Teichmann, Despina Moraitou, Sotirios Papagiannopoulos, Magda Tsolaki

**Affiliations:** 1Network Aging Research, Heidelberg University, Bergheimer Str. 20, 69115 Heidelberg, Germany; teichmann@nar.uni-heidelberg.de; 2School of Medicine, Aristotle University of Thessaloniki, University Campus, 54124 Thessaloniki, Greece; ubic2@otenet.gr (S.P.); tsolakim1@gmail.com (M.T.); 3School of Psychology, Aristotle University of Thessaloniki, University Campus, 54124 Thessaloniki, Greece; despinamorait@gmail.com; 43rd Department of Neurology, Papanikolaou General Hospital, Exohi, 57010 Thessaloniki, Greece; 51st Department of Νeurology, AHEPA University Hospital, University Campus, 54124 Thessaloniki, Greece

**Keywords:** education, personnel, workforce development, Alzheimer’s disease, health facilities, attitude, knowledge, confidence

## Abstract

People with Dementia (PwD) are frequently admitted in general hospitals. However, health care professionals have lack of dementia knowledge, negative attitudes toward dementia, and lack of confidence in caring those patients. The aim of this study is to develop, implement and evaluate a dementia staff training program in Greek general hospitals. It was a repeated-measures research design. Fourteen (14) two-day workshops were conducted, consisting of six targeted and interactive modules. Staff members (N = 242) attended the program and were assessed according to (1) individual performance: questionnaires about attitudes towards dementia, confidence in care, knowledge about dementia and anxiety before, immediately after the training and three months later, (2) an overall training evaluation immediately after the training and (3) an evaluation of training implementation three months later. Positive attitudes towards dementia, improvement of confidence in care and decrease of feeling of anxiety as a trait, were sustained over time. Knowledge about dementia also increased after the training and sustained, with, however, a slight decrease over time. A well applied training program seems to provide the basis of a better care in PwD during hospitalization. However, changes in the organizational level and a transformation of care culture are necessary for training sustainability over time.

## 1. Introduction

Due to an increased life expectancy, the world’s elderly population continues to grow rapidly, with the number of People with Dementia (PwD) rising in parallel [1]. PwD are more likely to be admitted to general hospitals than people of similar age without dementia [2]. Studies demonstrate that 12–76% of PwD use the general hospital beds [3,4,5,6] almost 2 times per year [6]. Moreover, PwD stay in hospital longer [7] being vulnerable to poor outcomes [2], and contributing additionally to an increased health cost [8,9,10]. Poor nutrition and hydration, falls, loss of functionality, increased disorientation, delirium, frequent use of antipsychotic drugs, hospital infections, adverse responses to medications and a high mortality rate are some of the consequences of hospital admission [11,12,13,14]. Up to 75% of PwD admitted to an acute hospital will experience Behavioral and Psychological Symptoms of Dementia (BPSD) [15] that cause further difficulties and distress to the caregivers [16,17,18]. Moreover, the hospital surroundings appear to be frightening and confusing for PwD [19], as it can worsen their BPSDs [11]. Additionally, due to cognitive impairment or/and BPSDs, PwD are also at risk for abuse, neglect or care omissions [20,21,22].

Poor knowledge among healthcare professionals may lead to a delayed diagnosis and to misinterpretation of symptoms, resulting in inappropriate treatment or lost opportunities in treatment [23,24]. Moreover, poor communication skills, negative attitudes toward dementia, lack of confidence in caring [25] and ageism [26] further contribute to poor quality of PwD care [11,25,27]. These facts concluded that suboptimal care in hospital settings is due to the inadequacy of staff—especially nurses—to meet the needs of PwD in a person-centered way [28].

### 1.1. Existing Training Programs

Due to a significant increase in number of patients suffering from dementia, the global dementia plan aims to improve the awareness towards the disease and set as a target 75% of countries to develop national policies until 2025 [29]. The World Alzheimer Report 2016 provided important recommendations to strengthen healthcare systems [2]. Thus, dementia training for clinical staff in hospital settings is steadily increasing [11].

According to the literature, many training programs have already been conducted in general hospitals globally. Educational interventions among staff seemed to improve their general knowledge about dementia [30,31,32,33,34,35,36,37,38], their attitude towards dementia [34,37,39,40], self-efficacy or confidence in caring for PwD [30,31,32,33,34,35,37,39,40,41,42], reduce their stress relating to caring [43] or generally have a positive effect on staff. However, a recent review revealed that dementia awareness training alone will not improve dementia care or outcomes for patients with dementia. Instead, how staff are supported to implement learning and resources and adopt good dementia care is a key component for improving health care practices and patient outcomes [44]. But this is a complex procedure, due to the fact that multiple person, social and organizational factors may affect the training outcomes [27]. Moreover, the lack of long-term studies and methodological differences of the implemented programs make the comparison of outcomes difficult [45]. Thus, due to heterogeneity of these programs’ design, there was a need to focus on training evaluations by means of their effectivenes and sustainability. Two recent literature reviews and one narrative synthesis provided some characteristics of the most effective approaches of dementia training for hospital staff [45,46,47]. According to the evidence, the great variability in teaching methods [45,46], interdisciplinary and tailor made programs [46], experiential, interactive and practice-based learning [45,46,48,49] videos, group activities and scenario-based exercises [45,47,49], tools which can be directly applied in the clinical practice [45,47,48], Personal Centered Care (PCC) approaches [45], champions/mentors and generally supporting conditions in service experts [45,47,48], were among the most utilized and successful characteristics of training programs for employees. Further features that can be used in the design of successful training programs are: relevant content of training and learning activities for staff, face to face programs rather than online programs [45,47,48], at least one day training (ideally full day sessions), inclusion of PwD and/or their caregivers via direct involvement or by videos and vignettes [47]. Moreover, sessions relevant to general knowledge about dementia, communication, dementia care and managing challenging behaviors were the most frequent training topics [45].

Although Greece enacted the national dementia strategy in 2014 in which the education of professional caregivers in hospital settings is included (action 2, axis 7 for the Greek national plan), there is a lack of developing dementia training programs in nursing personnel.

### 1.2. Aim of the Study

Due to the high number of PwD admitted in general hospitals and additional lack of staff skills towards dementia care, the idea for this work came from a larger research project of Network Aging Research (NAR): “People with dementia in acute hospitals”, which aimed to improve the quality of hospital care for cognitively impaired patients.

Thus, the aim of the current study intervention is to develop, implement and evaluate a dementia staff training program in general hospitals. This is the first time a training program in hospitals was conducted in Greece. Our research hypotheses were that: (1) confidence in care, knowledge and attitudes toward dementia will be positively changed after the training (2) and these changes will be sustained over time. Moreover, (3) the state and trait of anxiety will be decreased after the training and (4) these changes will be sustained over time. Lastly, it is expected (5) that there will be a positive evaluation of the whole intervention, and also (6) the implementation of what staff has learned in their daily routine.

## 2. Materials and Methods

### 2.1. Developing the Training Protocol

To develop the training protocol, a series of actions followed (Figure 1):Before developing the training curriculum, a study to explore the needs of hospital personnel on an upcoming dementia training was conducted (May 2017–July 2019). In order to develop a successful training program which may effectively transfer the theoretical concepts to practice, an educational needs analysis was necessary. Its aim was to explore the current status, interests and expectations of nursing staff, head nurses and physicians regarding a dementia training program [50]. According to bibliography, by exploring the educational needs of the target group and making the staff part of the training process, their motivation to learn and the implementation of what they learned is being increased [51].A narrative synthesis of the effectiveness of dementia staff training programs (January 2017–June 2020), was conducted. Its aim was to give an overview of the existing and established training programs in general hospitals and to explore the effectiveness of each training program [45].A new curriculum of dementia training was developed by an author (M.G.) (October 2018–March 2019). It was based on the staff “educational needs” analysis, on the “effective trainings” study and two established trainings conducted in England (“Getting to know me”) and Greece (“Positive care in dementia-train the trainers”) [30,52]. The curriculum was developed after consultation meetings among the project lead (M.G.), a dementia clinical specialist (M.T.) and two specialists in gerontology (D.M. and B.T.). A series of 14 two-day training workshops were conducted and lasted from April 2019 to February 2020.At the same time a further action took place, engaging hospital managers (September 2018–September 2019). Meetings between the research team and the participating hospital sectors teams were conducted to exchange experiences and agree on the needed actions of holding the training workshops. Moreover, barriers regarding the hospital stays of PwD, limited staff, training issues and unmet needs such as proactive training and hospital-wide system modifications were discussed.As the last step of protocol development, some action plans (July 2020) were scheduled for future use. The goal of this initiation was to set a Train-the-trainer program after the main training by the developer: Hospitals will identify key staff to deliver dementia training locally twice per year. Workshops will be delivered by a qualified trainer, with nominees attending for two full days of coaching before being signed-off as competent. Setting action plans or goals for newly acquired knowledge, are important factors, to “transfer the design” of the program into the practice [51].

### 2.2. Study Design

A repeated-measures research design was used to evaluate the impact of a dementia education program on hospital staff participants. Both hospital settings were assigned to one condition: ‘’Intervention Group’’, in an attempt to control bias, by keeping the same participants, demographics and personal characteristics in multiple measures over the time. Outcome measures were administered: before the training (Time Point 1, T1), directly after the training (Time Point 2, T2) and 3 months follow up (Time Point 3, T3).

### 2.3. Settings and Participants

Hospital administrators of three of the largest metropolitan general hospitals of Thessaloniki-Greece were invited via phone calls or personal contacts by two authors (M.T. & M.G.) and two of them—a general hospital (Papanikolaou-General Hospital, site A) and a university hospital (AHEPA-University Hospital, site B)—agreed to participate in the dementia training program. After obtaining the official approval from hospital administrators and the hospital’s scientific committee to conduct the training, the nurse service director (K.S.) of site B and the manager of “quality research and continuing learning department” (C.C.) of site A invited the decision makers of each ward (head nurses) to participate via an e-mail or phone calls. Due to the fact that caring PwD could be taken place in every ward (because dementia is mostly a secondary diagnosis) and additionally, ward rotation is a usual way of working among hospital personnel, no exclusion criteria of ward recruitment or participants were applied. The sample size of the intervention was calculated taking into consideration the G*Power [53]. In order to detect an effect of η^2^
*p* = 0.04 with 80% power in the respective ANOVA (alpha = 0.05), G*Power suggests 79 participants in each time point (T1, T2, T3) (*n* = 237).

The recruitment was a random selection, keeping a priority, among those who were on duty that time and expressed their willingness to participate. Thus, 241 nursing staff, head nurses, physiotherapists, administrative staff and one patient transport service driver were recruited as participants working in several wards, laboratories or in administration sector.

### 2.4. Training Intervention

Taking into consideration the list of staff members who expressed the willingness to participate and keeping a priority, 14 groups were formed. The intervention included classroom/face to face teaching, consisting of 6 targeted and interactive modules (two-day workshops) (Figure 2), each of 70–100 min duration (over 9 h in total). Program was delivered to staff in interdepartmental groups of 13 to 20 persons, by one of the authors (M.G.). Evaluation was conducted before, immediately after the training and 3 months later (T1, T2, T3), each lasting 20–30 min.

### 2.5. Ethical Standards

All procedures contributing to this work comply with the ethical standards outlined in the declaration of Helsinki, which is relevant to the national and institutional committees on human experimentation. The study was approved by the scientific committees of the two participating hospitals (site A: 1505/18th/ 26.9. 2018, site B: 12196/19th/ 26.7. 2018) and the research ethics committee of the Aristotle University of Thessaloniki (AUTH) (3/2.5. 2018). Staff participated voluntarily in the study. They were informed about the aim of the intervention, and subsequently they provided their written consent for participation. The General Data Protection Regulation GDPR in a research context [54], and the Greek Law of Data Protection were respected through the confidentiality and anonymity of the data.

### 2.6. Measures-Evaluation of the Program

A mixed method approach was adapted. Quantitative (validated questionnaires) and qualitative data (open-ended questions) were assessed at three time points.

#### 2.6.1. Individual Evaluation

Participants completed the following evaluation material: (1) a pre-training form about demographics, (2) a pre-post-follow up (3 months) assessment via self-reported questionnaires: the Dementia Knowledge Assessment Tool 2 (DKAT2) [55], the Dementia Attitudes Scale (DAS) [56], the Confidence in Dementia Scale [30] (CODE) and the State-Trait Anxiety Inventory (STAI) [57].

##### Questionnaires

Dementia Attitudes Scale (DAS)—a 20 item questionnaire

The DAS measures attitudes toward dementia [56]. Regarding the Greek version, total Cronbach’s alpha coefficient for DAS-GR was good (Cronbach’s α = 0.74) [58]. Exploratory Factor Analysis revealed two factors: the first factor (α = 0.67) labeled as “dementia knowledge” consisting of statements like “People with ADRD can be creative”. The second factor (α = 0.72) labeled as “social comfort” including statements like “I feel confident around people with ADRD”. Response options were scored on a 7-point Likert scale ranging from 1 (strongly disagree) to 7 (strongly agree). Higher scores indicate more positive attitudes and six items were reverse scored.

Dementia Knowledge Assessment Tool (DKAT2)—a 21-item questionnaire

The “DΚAΤ2” was developed to measure care workers’ foundation level knowledge of dementia [55]. Regarding the Greek version, it is a questionnaire with marginally adequate reliability: Cronbach’s alpha = 0.68 [58]. Items included statements like: “Blood vessel disease can also cause dementia”, “Only older adults develop dementia”. Response options are Yes/No/Don’t know. Thirteen items are correct statements and eight items are incorrect, which were reverse scored. Right answers are rated with one (1) and wrong or don’t know answers with zero (0). Higher score indicates higher knowledge in dementia topic.

Confidence in Dementia Scale (CODE)—a 9-item questionnaire

The CODE was developed to measure hospital staff confidence working with PwD [30]. It is a uni-dimensional questionnaire and its internal consistency reliability in Greek version is very good (α = 0.85) [58]. Items included statements like: “I feel able to manage situations when a person with dementia becomes agitated”. Response options were scored on a 5-point Likert scale with anchored ratings of “not confident”, “somewhat confident”, and “very confident”, with a higher score representing better confidence in working with PwD. Cut-off points within the scale are as follows: 0–18 not confident, 19–35 somewhat confident, 36–45 very confident [31].

State-Trait Anxiety Inventory (STAI)—a 40-item questionnaire

The STAI measures the state and trait of anxiety, in adults [57]. State anxiety (S-Anxiety) refers to the subjective feeling of tension, nervousness, and worry at a given moment. Trait anxiety (T-Anxiety) refers to relatively stable individual differences in anxiety as a personality trait, indicating the tendency to perceive and respond to stressful situations providing the intensity of state anxiety (S-Anxiety) reactions [59]. Cronbach’s alpha of the Greek version was found to be 0.93 for the State and 0.92 for the Trait subscale [59] providing a reliable tool for anxiety evaluation. The S-Anxiety scale consists of twenty statements that evaluate how the respondent feels “right now, at this moment”. The T-Anxiety scale consists of twenty statements that evaluate how the respondent feels “generally”. Both scales response options are scored on a 4-point Likert describing the intensity of feelings from 1 (not at all) to 4 (very much) or from 1 (almost never) to 4 (almost always), respectively. Higher scores indicate high levels of anxiety while ten S-Anxiety items and six T-Anxiety items have a reverse score [59].

#### 2.6.2. Overall Training Evaluation

An evaluation, using Likert scales or open-ended questions about the content, the training environment, the trainer and how well it was organized, was conducted immediately after the end of the training. Participants were asked if it was the first training they ever attended, and whether they had ever cared for a PwD in the past. Moreover, participants were asked if they felt confident enough to transfer their knowledge to others, to evaluate the most useful module, and what they would change if they repeated the training. They were also free to give feedback with free comments/reflections relevant to the experience of training.

#### 2.6.3. Implementation of the Training in Daily Routine

In the follow up assessment (3 months later), apart from the questionnaires, participants were asked to answer open questions relevant to the implementation of the knowledge they acquired by the training. Five multiple choices and open-ended questions were assessed: if they had the chance to care a PwD in the past 3 months, to use any tool to detect dementia, delirium or pain, to use the ‘’getting to know me’’ card, to use any communication technique or person center approach, or if they had discussed the training with others.

### 2.7. Data Analysis

The statistical analysis was carried out using SPSS V. 25.0. Demographic characteristics and previous exposure to dementia training or previous experience in working with PwD were analyzed using descriptive statistics, reported as percentages for categorical variables. Repeated method one way ANOVAs were carried out on each of the scales or their subscales (DKAT2, DAS comfort, DAS knowledge, CODE, STAI-S, STAI-T) to establish whether there were significant differences between the responses on the questionnaires between T1 and T2 and between T1, T2 and T3. Bonferroni adjustment was used for the multiple comparisons. Missing data were handled by mean substitution, and thus missing cases were replaced by means of the variables, while cases with many missing values were excluded. To analyze and quantify open-ended questions, a thematic analysis was conducted and categories were built. Data were compiled, disassembled through coding, reassembled and interpreted. Descriptive statistics were used to report the percentages and means for these categorical variables.

## 3. Results

### 3.1. Demographics

The total number of staff who completed the training program was 242. However, participants with missing post measurements and follow ups were excluded from the analysis. Thus, 232 participants completed the measurements before and immediately after the training, and 103 completed the three months follow-up, respectively. Due to the COVID-19 pandemic, the protective protocol for the healthcare systems was followed, thus the research team couldn’t reach the participants in their wards and questionnaires distributed through a link in Google Drive. The attrition rate in the 3-month follow up was high (57%,) either because participants did not systematically use an e-mail or they did not fill out the questionnaires due to other unknown reasons. 48.8% of participants came from site A and the rest from site B. The majority of participants were nurses (90.9%), female (94.6%), from 46 to 55 years old (64.9%), employed in hospitals for over 15 years (81.4%). Most of the attendees (71.1%) had an undergraduate degree (16 years of education) and 21.9% of them had at least one master degree, while 25.6% had a vocational diploma or a professional certificate from a private or public school/college. Most participants (74%) came from 17 different wards including surgery clinics (general surgery, neurosurgery, cardio surgery, plastic surgery). Fewer participants employed in the administrative sector (7.8%), laboratories (7.9%), emergency unit (6.2%), and regular outpatient clinics (5.4%). The 88% of the sample reported that they had received no training in dementia care in the past and the 47.5% of attendees claimed that they had been working at least once with a PwD (Table 1).

#### 3.1.1. DAS

Pre-post

Firstly, a repeated measures ANOVA was conducted (*n* = 230) to search significant differences in staff attitudes (DAS comfort and knowledge) towards PwD between T1 and T2. There was a significant main effect of time on the DAS (comfort) score, F(1,229) = 79.302, *p* < 0.001, η^2^ = 0.257. Bonferroni adjustment was used to investigate the differences of the means between T1 and T2. The comparisons indicated that there was a significant difference between the average score at T1 (M = 42.60, SD = 8.163) and T2 (M = 46.63, SD = 7.866). Regarding the DAS (knowledge), there was also a significant main effect of time F(1,229) = 294.010, *p* < 0.001, η^2^ = 0.562. The comparisons using the Bonferroni test indicated a significant difference between T1 (M = 61.98, SD = 5.700) and T2 (M = 67.84, SD = 5.103), *p* < 0.001, suggesting a more positive attitude toward PwD after intervention.

Pre-post-follow up

To search whether there were significant differences in staff attitudes (DAS comfort and knowledge) at T1, T2 and T3, a repeated measures ANOVA was conducted (*n* = 103). In regards to DAS (comfort) the data was met with the assumption of sphericity (χ^2^(2) = 2.887, *p* > 0.05). There was a significant main effect of time on the DAS (comfort) score F(2,204) = 21.37, *p* < 0.001, η^2^ = 0.173. Bonferroni was used to investigate whether differences between the means occurred between T1 and T2, T2 and T3, or T1 and T3. The comparisons indicated that there was a significant difference between the average score at T1 (M = 43.18, SD = 8.455) and T2 (M = 47.01, SD = 7.032), *p* < 0.001. There was also a significant difference between T1 and T3 (M = 47.44, SD = 7.692), *p* < 0.001, but not between T2 and T3 (*p* = 1.000), suggesting improvements in attitude (factor: comfort) over time. Regarding the DAS (knowledge) the data violated assumptions of Mauchly’s test of sphericity (χ^2^(2) = 8.553, *p* < 0.05) meaning that there is an increased probability of a Type II error. Since the estimate of sphericity was <0.75 the adjusted Greenhouse–Geisser F ratio [60] was used to make the F-value more conservative. There was a significant main effect of time on the DAS (knowledge) score F(2,204) = 70.264, *p* < 0.001, η^2^ = 0.408. Bonferroni test was conducted to investigate the differences between T1 and T2, T2 and T3, or T1 and T3. The comparisons indicated a significant difference between T1 (M = 62.66, SD = 5.333) and T2 (M = 68.39, SD = 4.262), *p* < 0.001. There was also a difference between T1 and T3 (M = 67.44, SD = 5.143), *p* < 0.001, but the difference between T2 and T3 was not significant (*p* = 0.212), suggesting that improvements in attitude (factor: knowledge) sustain over time (Figure 3, Table 2).

#### 3.1.2. CODE

Pre-post

A repeated measures ANOVA was used, (*n* = 232), to measure the confidence of caring PwD (CODE) at T1 and T2. A significant main effect on time was observed in CODE score, F(1,231) = 325.46, *p* < 0.001, η^2^ = 0.585. Bonferroni test was applied, and T1 score (M = 23.86, SD = 6.839) was lower than T2 (M = 31.31, SD = 6.083), *p* < 0.001, suggesting a higher feeling of confidence immediately after the intervention.

Pre-post-follow up

Another repeated measures ANOVA was conducted in staff confidence of caring PwD (CODE) at T1, T2 and T3. The data (*n* = 102) was met the assumption of shpericity (χ^2^(2) = 0.203, *p* > 0.05) and a significant main effect on time was observed in CODE score F(2,200) = 114.38, *p* < 0.001, η^2^ = 0.534. Bonferroni test was applied and T1 (M = 23.77, SD = 6.839) was lower than T2 (M = 31.43, SD = 6.083), *p* < 0.001. There was also significant difference between T1 and T3 (M = 32.31, SD = 5.767), *p* < 0.001 and there was not any difference between T2 and T3 (*p* = 0.457), suggesting improvements in confidence over time (Figure 4, Table 2).

#### 3.1.3. STAI

Pre-post

To measure the staff’s State of Anxiety (STAI-S) and Trait of Anxiety (STAI-T) at T1 and T2, a repeated measures ANOVA was conducted. Regarding the STAI-S data (*n* = 229) a significant main effect on time was observed F(1,228) = 23.382, *p* < 0.001, η^2^ = 0.093. Bonferroni test was applied and the comparisons of means indicated differences before and after the training while T1 score (M = 34.10, SD = 10.960) was higher than T2 (M = 30.87, SD = 10.285), *p* < 0.001. The STAI-T data (*n* = 229) showed a significant effect on time, too, F(1,228) = 28.393, *p* < 0.001, η^2^ = 0.111, with the comparisons of means to indicate differences between T1 score (M = 42.06, SD = 6.882) and T2 (M = 40.36, SD = 6.990), *p* < 0.001, suggesting a decrease in anxiety as a state and as a trait immediately after the training.

Pre-post-follow up

Further repeated measures ANOVAs tested the staff’s State of Anxiety (STAI-S) and Trait of Anxiety (STAI-T) at three time points (T1, T2, T3). The STAI-S data (*n* = 101) was met with the assumption of sphericity (χ^2^(2) = 2.674, *p* > 0.05) and a significant main effect on time was observed F(2,200) = 10.52, *p* < 0.001, η^2^ = 0.095. The comparisons of means indicated differences between three time points, and T1 score (M = 33.24, SD = 10.440) was higher than T2 (M = 29.51, SD = 9.546), *p* < 0.001. There was also a significant difference between T2 and T3 (M = 34.02, SD = 11.378), *p* < 0.001, but there was no difference between T1 and T3 (*p* = 1.000), suggesting that the state of anxiety improved only after the intervention, but was not sustainable over time. Regarding the STAI–T (*n* = 100), again the assumption of sphericity (χ^2^(2) = 2.275, *p* > 0.05) was met and there was a main effect on time F(2,198) = 5.68, *p* = 0.004, η^2^ = 0.054. The means occurred differences between three time points and the average score was higher at T1 (M = 40.82, SD = 7.353) than T2 (M = 39.25, SD = 7.237), *p* = 0.005. There was also a significant difference between T1 and T3 (M = 39.48, SD = 7.876), *p* = 0.044, but no difference between T2 and T3 (*p* = 1.000), indicating that Anxiety as a trait decreased immediately after the intervention and was generally sustained over time (Figure 5, Table 2).

#### 3.1.4. DKAT2

Pre-post

Regarding the general knowledge about PwD (DKAT2) two ANOVAs were conducted about the correct and don’t know answers, respectively, in T1 and T2. The analysis of DKAT2 (correct answers) scores (*n* = 231) showed a significant effect of time, F(1,230) = 500.21, *p* < 0.001, η^2^ = 0.685. Bonferroni test applied and the comparisons between two time points revealed a significant difference between the average score, which was lower at T1 (M = 12.23, SD = 2.977) than T2 (M = 16.89, SD = 1.756), *p* < 0.001. Regarding the DKAT2 (don’t know answers) (*n* = 231), there was also an effect of time F(1,230) = 341.89, *p* < 0.001, η^2^ = 0.598., and T1 (M = 5.15, SD = 3.841) was higher than T2 (M = 0.67, SD = 1.287), *p* = 0.001, suggesting that correct answers increased after the training while the don’t know answers decreased, respectively.

Pre-post-follow up

To search main differences in three time points, two further ANOVAS about the correct and don’t know answers were conducted. Specifically, the analysis of DKAT2 (correct answers) scores (*n* = 102) showed a violation of assumptions of Mauchly’s test of sphericity (χ^2^(2) = 36.991, *p* < 0.05) and the adjusted Greenhouse–Geisse F ratio was used. There was a significant main effect of time on the DKAT2 (correct answers) score F(2,202) = 179.83, *p* < 0.001, η^2^ = 0.640. The comparisons between three time points revealed a significant difference between the average score, which was lower at T1 (M = 12.33, SD = 3.173) than T2 (M = 17.24, SD = 1.517), *p* < 0.001. There was also a significant difference between T1 and T3 (M = 16.13, SD = 1.938), *p* < 0.001 and between T2 and T3 (*p* < 0.001), suggesting improvements in general knowledge about dementia over time, with a slight decrease in the follow up measurement. DKAT2 (don’t know answers) (*n* = 102) also violated assumptions of Mauchly’s test of sphericity (χ^2^(2) = 122,774, *p* < 0.05) and the adjusted Greenhouse–Geisse F ratio was used, F(2,202) = 133.48, *p* < 0.001, η^2^ = 0.569 revealing a significant main effect of time. T1 (M = 5.21, SD = 4.215) was higher than T2 (M = 0.50, SD = 1.022), *p* < 0.001. There was also significant difference between T1 and T3 (M = 1.09, SD = 1.561), *p* < 0.001 and between T2 and T3 (*p* < 0.001), suggesting a decrease of don’t know answers in the staff’s general knowledge over time, with a slight increase after 3 months. Generally, these findings suggest a significant improvement in gaining knowledge over time; however, some of the knowledge got lost (Figure 6, Table 2).

### 3.2. Overall Training Evaluation

Of the 234 participants who completed the evaluation immediately after the intervention, 85% described the training as excellent and they felt quite confident to distribute the acquired knowledge to others (mean = 8.2/10). The most useful modules were three: “Understanding and dealing with challenging behavior” (36%), “Communication” (14.9%) and “Care of dementia–practices in activities of daily living” (25.4%), while an important percentage (27.2%) reported that all six modules were useful. Besides, the way in how the training was organized, tools/material and the trainer performance were also evaluated as excellent by the majority of participants. Only the physical environment was evaluated lower (Table 3).

Sixty-two out of 234 participants (26.5%) provided written comments about what they would change if the training was repeated. Thus, 47 participants (75.8%) claimed that if they had the chance to change the delivery and content of training, they would add more videos with PwD, preferable videos from Greek actors/fit to Greek reality, more scenarios and role playing, and they would eliminate the written exercises. Moreover, they suggested a more detailed section about family caregivers training and how to make them cooperate during hospitalization, and another section about the systematic physical training for the patients. Besides, few of them would like to learn more details about the stages and the types of dementia. Five participants (8.1%) suggested further actions such as the participation of a PwD group or the opportunity to visit them, or action plans like identifying and writing down the number of PwD in each ward. Some of the participants claimed that it would be necessary to train all hospital staff. Thirteen participants (21%) commented for the duration of the training. Most of them claimed that the duration of the training or the discussion should be longer, while few suggested minimizing training hours. Another three participants (4.8%) commented that the physical environment disturbances such as the illumination and high temperature of the room.

Forty-three out of 234 participants (18.4%) provided written feedback about the overall training experience. Many of them (*n* = 19, 44.2%) gave general positive feedback about the training: how useful it was to provide a treatment protocol for PwD to each ward and the idea that all professionals should be trained. Many attendees (*n* = 17, 39.5%) commented also positively on the delivery and the content of training. Specifically, they highlightened the innovative, creative, impactful and educative character of training through videos, interesting topics, group interaction and use of many examples for better understanding. Eleven participants (25.6%) gave positive feedback to the trainer, and another six (14%) liked the learned tools and techniques. Few participants expressed their doubts (*n* = 5, 11.6%) on how this knowledge can be implemented in the Greek hospital reality, taking into consideration the lack of time during the shift and the lack of staff members.

### 3.3. Implementation of the Training in Daily Routine

One hundred and three participants completed the follow up, answering in further 5 open-ended and multiple choice questions relevant to the implementation of the acquired knowledge in their daily routine. 45.1% of participants (*n* = 46) worked with a PwD in the last three months and only 14% (*n* = 14) used at least one tool to detect dementia, delirium or pain. While few staff used the “getting to know me card” (*n* = 8, 8.8%) at the patients’ bedside, many of them used techniques of communication (*n* = 45, 50%) and specifically person-centered practices (*n* = 20, 22%) to deal with the patients during hospitalization. “Verbal and non-verbal communication” such as calm tone of voice, eye contact, smile and kindness, friendly body position/contact, kind touch, appropriate way of speaking—including repetitions and orientation in hospital environment—were the most frequent answers. Other techniques based on “PCC approach” were also used during the last three months. By using photos or talking about a favorite routine as a motivation to discuss, participants claimed that they tried to trigger pleasant memories for the patients. They also tried to give time, to show that they were interested, and to find out patients’ needs in order to avoid anxious situations for them. Besides, techniques of grooming and eating were referred to as useful, being supportive and focusing on the strength of patients and eliminating their losses. Getting information from family caregivers and asking them to stay with the PwD was also suggested from the attendees in the last three months. Besides, 48% of participants (*n* = 48) discussed the training with other persons. Specifically, 23.2% (*n* = 23) discussed the experience of training with their colleagues, either with those who had attended the program (*n* = 16, 16.2%) or those who had not attended the program (14.1%, *n* = 14). Few of them (*n* = 7, 7.1%) discussed it with family members (Figure 7).

## 4. Discussion

The purpose of the current study was to develop, implement and evaluate a dementia training program in hospital staff caring PwD. According to the main results, the effects of training on participants’ attitudes toward dementia, confidence in care and trait of anxiety were remarkable and sustained over time. Knowledge about dementia also increased after the training and sustained with a slight decrease over the time. The training was positively evaluated by the participants who also, highlighted its innovative and educative character. They applied the acquired knowledge in clinical practice by using mostly communication techniques and PCC approaches in daily care of Pwd. Main results are further discussed according to Holton’s three level evaluation model: learning outcomes, individual performance and organizational changes [51].

### 4.1. Learning, Individual Performance and Organization Outcomes

The current training was a two-day face-to-face workshop, based on interactive and varied teaching methods including PCC, creative exercises, videos and enthusiastic facilitator being supported by hospital management, characteristics which are among the most effective in dementia trainings [45,47,48,49]. Staff educational needs and expectations were explored before developing the curriculum [50]. Moreover, “participants’ readiness” for the training and “fulfillment of training content”, as motivation factors for successful trainings [45,61], had also been investigated prior to the intervention [50].

Learning outcomes were observed with a statistically significant improvement in reported knowledge, confidence in caring, attitudes toward dementia and feeling of State and Trait of Anxiety, immediately after the training. It has been suggested that reports immediately after the training may be less strong to capture the changes, instead, only pre-post and follow-up measurements can directly evaluate real changes [28,42]. Thus, concerning the comparisons revealed from the pre-post-follow up measurements, positive attitudes toward dementia, improvement of confidence in caring and decrease of feeling of anxiety as a trait, sustained over time. Anxiety as a state, improved only after the intervention, but was not sustainable over time. This finding is in accordance with another study revealed that the trait test is more stable than the state test across time [62]. A possible explanation is that, according to the multidimensionality of state and trait of anxiety, the “cognitive-worry” component of the state did not sustain, probably because it was not either a physical threatening situation [63] or a permanent situation of anxiety, while the facet of “daily routines situation” of trait did change and sustain over time [64]. Moreover, knowledge about dementia generally sustained over time, but a slight decrease was observed between the post and the delayed post measurement (Τ2: T3), increasing at the same time as the “don’t know answers”. These findings suggest that knowledge is improved after the training but when time passed, some of the knowledge is getting lost. Comparing our results with other studies, knowledge improvement is the most common learning outcome [30,31,34,35,36,37]. In some cases, similar interventions, with a pre-post or delayed post design indicated a decrease of knowledge after 4 months or 8 days [32,33]. Different education background, different work experience, high attrition rate or small sample, no validated measures, variations in the way individuals implement strategies, and motivation to attend or the obligatory attendance of trainings in some cases may have influenced the sustainability of these learning changes [32,42,65,66]. Level of knowledge is argued to be linked to confidence, as successful learners are expected to feel confident and therefore more motivated to transfer the acquired knowledge [51]. Furthermore, level of knowledge has also an influence in trainee’s attitude [67] and feelings of stress [43]. Thus, the improvement of knowledge leads to positive attitudes towards dementia, confidence increase and reduction of stress related to care, findings that are confirmed from other studies as well [28,30,31,32,33,34,35,37,39,40,41,42,43].

Regarding the individual performance, the use of communication techniques through PCC approaches, the use of tools to detect dementia, pain and delirium, or gathering patient information through documentation tools were all reported as changes in clinical practice three months later, being consistent with other study findings [28,30,31,32,35,36,39,40,41,42]. Specifically, calmness and kindness, friendly body position/touch, repetitions and orientation in a hospital environment were the most frequent communication techniques. Using photos, triggering pleasant memories or starting a conversation about a favorite routine, giving time to answer, getting information from family asking also them to stay with the patient, using techniques of grooming and eating based on patient’s strengths and eliminating their losses, were the most frequent PCC approaches. According to a recent study, the individualized care practices or use of PCC is frequently used when staffs knowledge and attitude are improved, a finding that is in line with this current study [68]. However, without institutional-level changes (care cultures that prioritized tasks, routines and physical health), hospital staff are often unable to provide PCC even when they have the experience and knowledge to do so [69].

Learning and individual performance changes are effective only if linked with organizational goals and evaluated across the organizational environment according to Holton’s model [61]. Setting action plans by means of organization initiations, such as the delivery of the program to all clinical managers and to all acute sites of hospitals, and repetition of staff training twice a year, are among the goals of the current study and are in line with other studies [42]. Moreover, creating champions or training the trainers are also effective goals of training [34,40,41] and this is another action plan of the current study at the organizational level. Champions or mentors are important for the transfer of knowledge because they teach, support and lead changes in the workplace [34], provide the relationships between services [40] and contribute to overall training sustainability [41].

### 4.2. Training Evaluation—Reflections

While the program evaluation was overwhelmingly positive regarding the trainer performance, the content, given tools and how it was organized, the physical environment where the training took place was evaluated as lower. Positive feedback regarding the entire training experience was given, highlightening the innovating, useful, creative and educative character of the training. Specifically, they positively commented on the group interaction, videos to present the voice of PwD and carers, better understanding due to many examples, learning of techniques and tools, findings that are in line with literature relevant to the design of successful training [45,46,47,48,50]. However, an interesting finding was regarding the doubts of some participants expressed on how this knowledge can be implemented in the Greek hospital reality, taking into consideration the lack of time during the shift and lack of staff members. Due to the financial crisis in Greece, there is a lack of financial resources in the health care system and additionally a serious shortage in staff members [70,71], while hospitals operate with one third of what National Nurses United stipulates to be the lowest safe staffing level [72]. Moreover, the average age of attendees in current study was 50 years old, working >15 years in hospitals, a finding that is line with the suspension of hiring staff due to crisis. Existing staff will age, making professional caregivers scarce and tasks more strenuous for them [71]. Thus, except from limited time, staff and financial resources, the unpredictable nature of the workload in acute care appeared to be the barriers to sustainable changes [73,74] increasing also their stress. Problems between the health care team (disagreements, miscommunication and violent episodes) could further increase the stress related to nursing job according to a recent study in Greek hospitals [75]. Other barriers could be the lack of practical support [70], team meetings and supervision and the accountability to the law, since the law is not clear concerning Greek nurses’ intervention framework. Nursing continues to be dependent on the medical profession’s authority without a clear autonomy/decision making [75].

Regarding the content of training modules, “Understanding and dealing with challenging behavior”, “Communication” and “Care of dementia–practices in activities of daily living”, commented as the most useful modules, a finding that is in accordance with the most common and the most interesting topics used in dementia trainings [45,50]. If they were to repeat the training, participants would add more videos with PwD, preferably fitting to Greek reality, more scenarios, role playing and more details about dementia theoretical background. It is worth noting that some participants suggested incorporating family caregivers in caring during hospitalization and they additionally recommended the participation of a PwD group, which is in line with practices of other studies by means of a “whole system approach” [76] linking all the persons concerned PwDs and their families [28,34,40]. The integrated caregiver network moderated by a health professional is needed to enable family caregivers to share learning experiences and enhance social support [77]. Duration of the training was also commented on by the participants; however, >8 h of total duration seems to be the most effective duration according to recent literature [48].

To summarize, a training program success depends on staff ability to put their learning into practice through behavioral change [78]. However, sustainability of good results or changes in individual behavior is more than a good implementation. Thus, changes in the organizational level, realigning the hospital strategy and a transformation of care culture are necessary for training sustainability over time. Specifically, the supporting managers or conditions in the frame of strong leadership, initiative approaches, teamwork and continuous action and redesigning the ward environment are the most valuable factors [30,49,61,65,78]. However, there are some objective barriers that make the organization changes difficult, like bureaucracy [74,79], hierarchical culture focused on internal stability, models of working resistant to flexibility/openness [80] and lack of resources (time, financial, staffing and environmental) [49,81], as already mentioned. Another objective barrier regarding to the Greek community, is poor distribution of existing healthcare resources due to the unique characteristics of the healthcare system. Culture of the healthcare system is hospital-centered, and the primary care is almost missing, resulting in overburdened hospitals. The need for development of the primary healthcare is crucial, in order to increase people’s access to care, and reduce overall costs [70,71]. The additional shortages of healthcare professionals have aggravated this specific problem [70].

Changing practices into an organization is a challenge. In recent research about the obstacles and perpective of changes in Greek hospitals, the view of change in the hospital sector was positively correlated to the idea of educational programs. Employees should be motivated to participate in change, while communication and better relations with colleagues were also stressed to contribute in change. In contrast, the obstacles of the adoption of a change is the lack of education/training, the lack of guidance, leadership as already mention and clear procedures. Administrative support is necessary for a change-oriented organizational culture environment [82]. No matter how good a training is, without continued in service training and hospital-wide transformation, the changes achieved will be short lived [32]. The complexity of improving dementia care in hospitals shows that a system-wide approach is needed [78].

## 5. Conclusions

This was the first training for staff caring PwD in general hospitals of Greece. Its aim was not only to enhance the knowledge, confidence in care and positive attitude towards dementia as most trainings do globally, but also to reduce stress in professional caregivers. The current training content was adapted to the unique characteristics of the Greek healthcare system with enormous staff shortages and nurse work burden, provided specific techniques on how to cope with stress related to the job.

Training evaluation is a crucial part of good implementation and sustainability of changes. Although there are some reviews, in the bibliography, which evaluated the trainings according to specific models, like Kirkpatrick’s [46,47], this is the first intervention, which was developed following the Holton’s evaluation model. Recognizing the complexity of training in environments such as hospitals is difficult. Thus, Holton’s model is a comprehensive framework, including an independent and highly relevant “organizational level” as an additional outcome. This model integrates not only training evaluation like the others, but also training effectiveness [83].

According to the current results, learning outcomes and individual changes in clinical practice were reported, while future action plans in organization level were set. Thus, conducting the training twice per year, more and more staff would be aware in dementia care issues, leading to care culture changes. The role of leadership and support of staff in evidence-based care of older people is essential.

To reach sustainable changes in clinical practice, the well trained and qualified employees are not enough; multi-level evaluations among hospital wards, but also across the organization are needed. Further research is needed with larger sample sizes including more settings in terms of implementation, sustainability, organization and patient outcomes. Moreover, as nursing teams work with other healthcare professionals in providing care for older persons in hospital, future studies should include interprofessional teams. Furthermore, dementia education should be transferred not only across the hospital sector but also at academic level, such as the nursing schools of universities. The gaps in the dementia care education of health care staff require strategic and integrated action adapted in each country according to the health system characteristics. The systematic cooperation between PwD and their families, researchers, academia, policy makers and individual practitioners is required to reach the best care for PwD. No-one alone can make the change towards quality care for PwD. Lastly, given the fact that in current study, staff’s trait of anxiety reduced, future research exploring the quality of care impact on the PwDs life, their families and health caregivers is needed.

## 6. Limitations

The current study has some limitations to report. The sample is not representative for the actual range of professions working in a hospital. Both quantitative and qualitative results relied on self-reports. However, mixed method measurements are more effective because quantitative measurements alone are not always sensitive to changes [84] and qualitative measurements through reflections are also valuable [85]. This was not a Randomized Control Trial (RCT) study, but there is still a debate on whether RCTs should be considered the “gold standard” in health education research [86,87] because the complexity of healthcare settings may demand simpler solutions or other types of randomization [88]. Due to Covid-19, it wasn’t possible to catch each participant for the follow-up, and thus the response rate was low (44.4%, 103 respondents out of 232). According to literature, the reported response rate for mailed surveys to the general population approach is 60%, whereas response rates for health professionals vary from 11% to 90% [89], a finding that is in accordance to our study. Moreover, the study design did not include patient outcomes.

## Figures and Tables

**Figure 1 brainsci-10-00976-f001:**
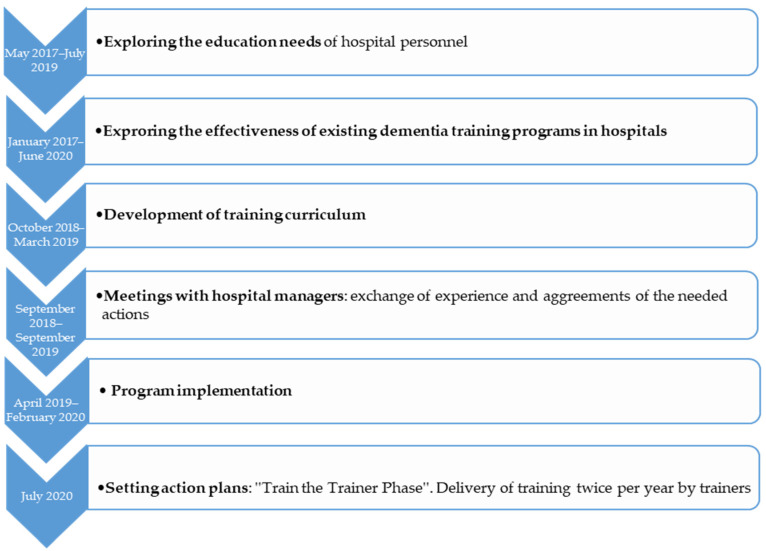
Overview of training protocol development.

**Figure 2 brainsci-10-00976-f002:**
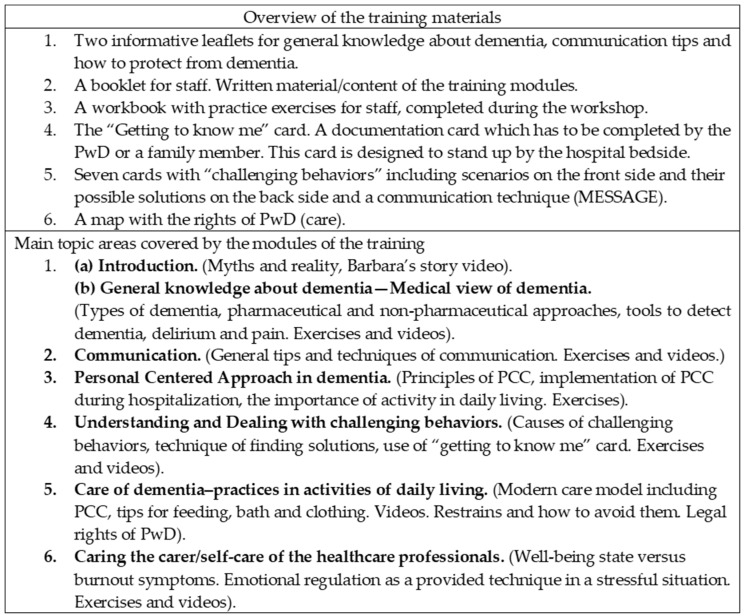
Overview of the training program and its materials/tools.

**Figure 3 brainsci-10-00976-f003:**
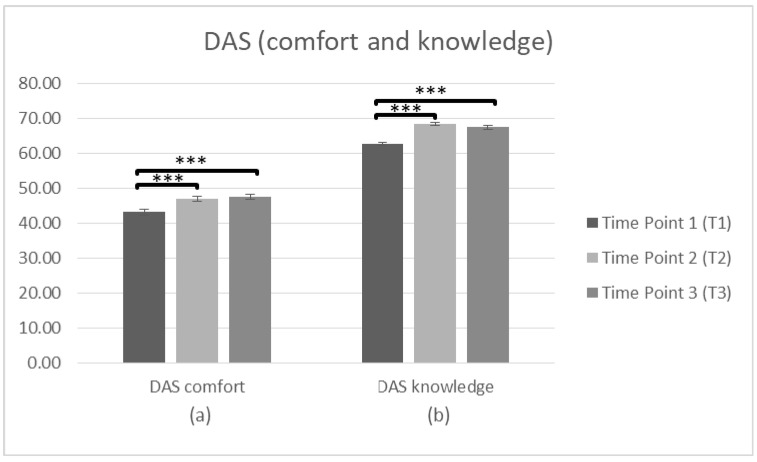
Changes in participants’ attitudes toward dementia (DAS) immediately prior to the intervention, immediately after the intervention, and after 3 months following the intervention for complete cases. Graph depicts the estimated effects for factors: (**a**) “DAS comfort” and (**b**) “DAS knowledge” (*** *p* < 0.001). Means are shown with error bars depicting standard error of the mean.

**Figure 4 brainsci-10-00976-f004:**
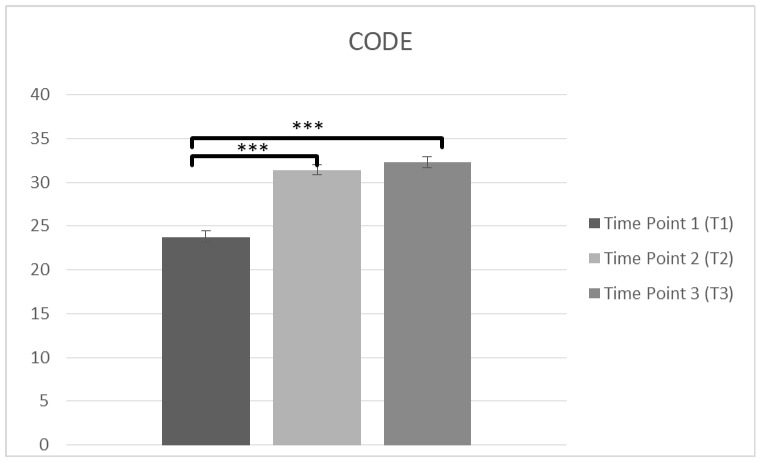
Changes in participants confidence in care (CODE) immediately prior to the intervention, immediately after the intervention, and after 3 months following the intervention for complete cases (*** *p* < 0.001). Means are shown with error bars depicting standard error of the mean.

**Figure 5 brainsci-10-00976-f005:**
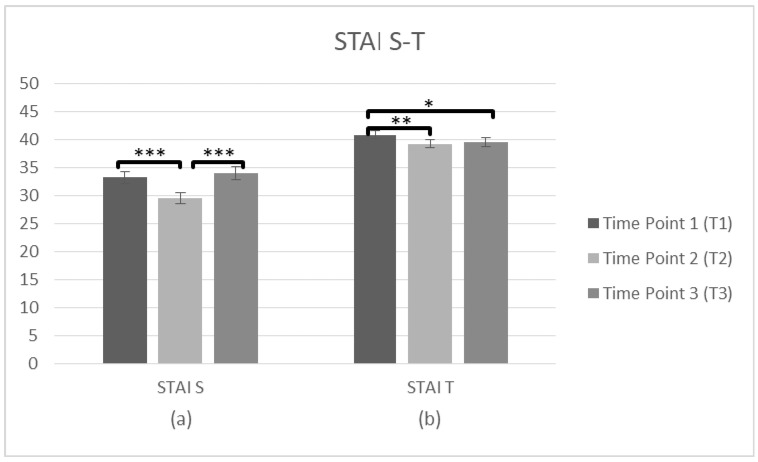
Changes in participants anxiety as a state and trait (STAI) immediately prior to the intervention, immediately after the intervention, and after 3 months following the intervention for complete cases. Graph depicts the estimated effects for (**a**) STAI S and (**b**) STAI T (* *p* < 0.05, ** *p* < 0.01, *** *p* < 0.001). Means are shown with error bars depicting standard error of the mean.

**Figure 6 brainsci-10-00976-f006:**
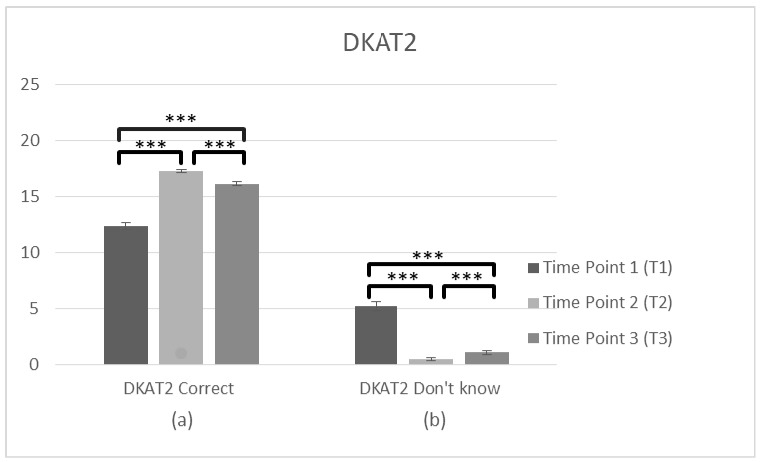
Changes in participant’s knowledge about dementia (DKAT2) immediately prior to the intervention, immediately after the intervention, and after 3 months following the intervention for complete cases. Graph depicts the estimated effects of (**a**) correct answers in DKAT2 statements and (**b**) don’t know answers (*** *p* < 0.001). Means are shown with error bars depicting standard error of the mean.

**Figure 7 brainsci-10-00976-f007:**
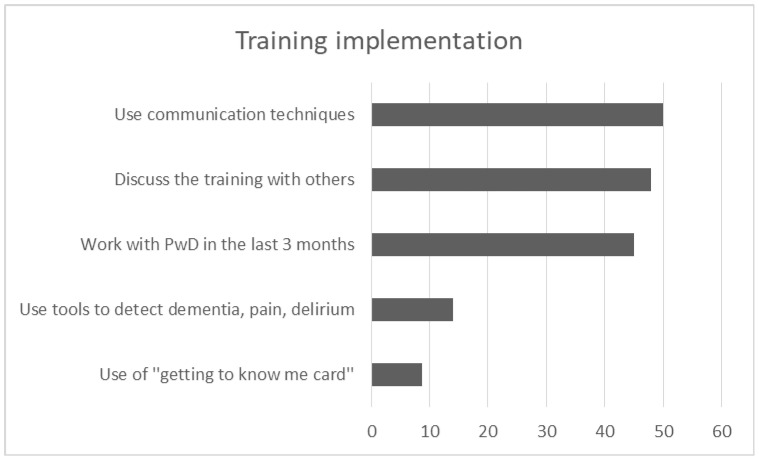
Graph depicts the most frequent answers in 5 open-ended questions of training implementation.

**Table 1 brainsci-10-00976-t001:** Participants’ demographics.

Demographic Characteristics	*n* = 242 (%)	Demographic Characteristics		*n* = 242 (%)
**Hospital settings** **Gender**	Site A: 118 (48.8) Site B: 124 (51.2) Female: 229 (94.6) Male: 13 (5.4)	**Years working in hospital setting**		1–5: 6 (2.5) 6–10: 9 (3.7) 11–15: 22 (9.1) >15: 197 (81.4) Missing: 8 (3.3)
**Age**	15–25: 1 (0.4) 26–35: 4 (1.7) 36–45: 53 (21.9) 46–55: 157 (64.9) 56–65: 20 (8.3) Missing: 7 (2.8)	**Previous dementia training**		Yes: 21 (8.7) No: 213 (88) Missing: 8 (3.3)
**Profession**	Nurses: 220 (90.9) Administrative staff: 10 (4.1) Physiotherapists: 6 (2.5) Patient transporter: 1 (0.4) Missing: 5 (2)	**Working with Pwd**		Yes:115 (47.5) No:119 (49.2) Missing: 8 (3.3)
**Education**	Undergraduate degree (university, 4 years): 172 (71.1) Vocational diplomas from schools/colleges other than university (2–3 years): 62 (25.6) Missing: 8 (3.3) Further education: One master: 48 (19.8) Two or more masters: 5 (2.1) Phd: 3 (1.2) Further university bachelors: 8 (3.3) Missing: 8 (3.3)			
**Participating wards**	Cardiology: 25 (10.4) Neurology: 15 (6.2) Psychiatric: 13 (5.4) Pulmonary: 7 (2.9) Artificial-kidney: 8 (3.3) Internal medicine: 14 (5.7) Hematology: 8 (3.3) Otorhinolaryngology: 4 (1.7) Emergencies unit: 15 (6.2)		Orthopedics: 2 (0.8) Gastroenterology: 4 (1.7) Ophthalmology: 1 (0.4) Surgery clinics (5): 56 (23.1) Intensive care: 9 (3.7) Regular Outpatient clinic: 13 (5.4) Laboratories: 19 (7.9) Administrative sector: 19 (7.8) Missing: 10 (4.1)	

**Table 2 brainsci-10-00976-t002:** Mean scores on staff Attitude, Confidence, State and Trait of Anxiety and Knowledge scales and within subjects ANOVA.

	Time1					Time2					Time3				
	*n*	Mean	SE	95%CI		*n*	Mean	SE	95%CI		*n*	Mean	SE	95%CI	
				Low	High				Low	High				Low	High
DAS (comfort)	103	43.18	0.83	41.52	44.82	103	47.03	0.69	45.66	48.40	103	47.44	0.76	45.93	48.94
DAS (knowledge)	103	62.66	0.53	61.62	63.70	103	68.39	0.42	67.56	69.22	103	67.44	0.51	66.43	68.44
CODE	101	23.77	0.68	22.42	25.12	101	31.43	0.61	30.23	32.63	101	32.31	0.57	31.17	33.45
STAI-S	101	33.24	1.04	31.18	35.30	101	29.51	0.95	27.62	31.39	101	34.02	1.13	31.77	36.27
STAI-T	100	40.82	0.74	39.36	42.28	100	39.25	0.72	37.81	40.69	100	39.48	0.79	37.92	41.04
DKAT2 (Correct)	101	12.33	0.31	11.71	12.96	101	17.24	0.15	16.94	17.53	101	16.13	0.19	15.75	16.51
DKAT2 (Don’t know)	102	5.21	0.42	4.38	6.03	102	0.50	0.10	0.30	0.70	102	1.09	0.16	0.78	1.40

**Table 3 brainsci-10-00976-t003:** Training evaluation.

Modules Evaluation (%)*n* = 234	Confidence in Distribute the Knowledge (Mean)*n* = 232	Most Useful Module (%)*n* = 228	Less Useful Module (%)*n* = 228
Module 1	Excellent: 80.3Very good: 16.3Fair: 3.4	7.9	6.6	13.2
Module 2	Excellent: 82.4Very good: 15.5Fair: 2.1	8.1	14.9	0.4
Module 3	Excellent: 79.4Very good: 20.6	8.1	8.3	3.5
Module 4	Excellent: 82.9Very good: 15.8Fair: 1.3	8.2	36	0.4
Module 5	Excellent: 81.2Very good: 17.1Fair: 1.7	8.3	25.4	0.4
Module 6	Excellent: 81.2Very good: 17.5Fair: 1.3	8.3	14.5	1.8
All modules	8.2	27.2	
**Evaluation-Total training (%)** ***n* = 234**	Excellent: 85Very good: 14.5Fair: 0.4	**Trainer evaluation (%)** ***n* = 234**	Excellent: 93.4Very good: 6.4	**How well was organized (%)** ***n* = 234**	Excellent: 81.6Very good: 18.4
**Material (%)** ***n* = 234**	Excellent: 77.8Very good: 19.7Fair: 2.6	**Training environment (%)** ***n* = 234**	Excellent: 62.8Very good: 33.3Fair: 3.4Bad: 0.4

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
