# Peer review of "Effects of a Person Centered Dementia Training Program in Greek Hospital Staff—Implementation and Evaluation"

_brainsci, 2020, doi:10.3390/brainsci10120976_

Round 1
Reviewer 1 Report
The manuscript titled as “Effects of a Person Centered Dementia Training Program in Greek Hospital Staff – Implementation and Evaluation” by Gkioka et al., aimed to develop, implement and evaluate a dementia staff training program in Greek general hospitals. However, health care professionals have lack of dementia knowledge, negative attitudes toward dementia, and lack of confidence in caring those patients. The draft needs minor speech editing. The data is well presented, but sometimes it is confusing because of too long phrases. Overall, the study is well designed and the results obtained appear to support the authors' conclusions.
However, there are a few issues that should be addressed:
I think subchapter 2.1. Developing the Training Protocol of the Materials and Methods should be rewritten; it is quite confusing and difficult to follow.
The names of the universities and / or hospitals involved in the study should be provided.
Author Response
Response to Reviewer 1 Comments
Point 1: The manuscript titled as “Effects of a Person Centered Dementia Training Program in Greek Hospital Staff – Implementation and Evaluation” by Gkioka et al., aimed to develop, implement and evaluate a dementia staff training program in Greek general hospitals. However, health care professionals have lack of dementia knowledge, negative attitudes toward dementia, and lack of confidence in caring those patients. The draft needs minor speech editing. The data is well presented, but sometimes it is confusing because of too long phrases. Overall, the study is well designed and the results obtained appear to support the authors' conclusions.
Response 1:
We would like to thank the reviewer for the constructive feedback and thorough comments of this manuscript. We followed closely the suggestions in order to revise our document.
We agree that in some points the phrases were too long and for that reason we re-phrase by shortening some of them. You can find the corrections here:
Chapter 1.1. line 69, 76-85,
Chapter 3.2: line 19
Chapter 4.1: line 59, 84, 89
Chapter 4.2: line 120-123, 144-146
Point 2: I think subchapter 2.1. Developing the Training Protocol of the Materials and Methods should be rewritten; it is quite confusing and difficult to follow.
Response 2:
Thank you for the constructive comment. Subchapter 2.1 is now rewritten, revealing the chronological order of actions needed for the protocol development. One action (ethics approval), has now removed to the 2.5 chapter (ethical standard) while another action (study of psychometric properties of questionnaires) was deleted and it is only reported as reference in chapter 2.1.6 (Questionnaires), lines: 216, 225, 233. Moreover, we added figure 1 giving the overview of developing the protocol in 6 bullets. We now think that it is easier for the reader to follow.
Point 3:
The names of the universities and / or hospitals involved in the study should be provided.
Response 3:
Thank you for this comment. Our thought was to conceal the person’s names, hospitals and university for the blind reviewing. It is now added to the manuscript:
2.3 chapter: line 170-172
2.5 chapter: line 200
6. chapter: 235-239.
Reviewer 2 Report
A cumbersome article, without a strict logic of presentation. It is based only on enumeration and presentation of irrelevant situations. No statistics are made of the realities presented in the mathematical sense, of the highlighting of some functions and of their variables. The implementation of Training has been insisted on for a long time, in the context of the topic of the article, which was dementia, without deciding, at least formally, the optimal age for the disease, its possible corrections, brought by medication or other therapeutic procedures.
Although a similar work is used in the bibliography, the barriers and facilitators to implementing dementia education and training in health and social care services, in Greek hospitals and what peculiarities have been observed are not highlighted at all.
The investigation made is not of quality and does not present a novelty in the field, being a simple presentation of some data collected on various criteria.
Author Response
Response to Reviewer 2 Comments
Point 1: A cumbersome article, without a strict logic of presentation. It is based only on enumeration and presentation of irrelevant situations.
Response 1:
We would like to thank the reviewer 2 for attentively reading of our manuscript. We followed closely the suggestions in order to revise our document.
However, according to this first point, we think that the structure of presentation follows strictly the journal’s standard. Introduction, methods/material, results and discussion. Due to the nature of the article (intervention) we considered as important, firstly to present the procedure of protocol development and then the study design, setting and participants, measures and data analysis as subchapters of ‘’Methods’’. Regarding the ‘’Measures’’ subchapter we divided it to three sections according to the type and time points of measurements, a method that is used by other studies too [1,2]. Moreover, the presentation of data, interpretation of results and discussion support and actually answer to the research questions reported in chapter 1.2 (aim of study).
Point 2: No statistics are made of the realities presented in the mathematical sense, of the highlighting of some functions and of their variables.
Response 2: Thank you for this comment. The statistics based on the ANOVAs tests for the quantitative data, highlightening the differences between times points. Qualitative data were explored by frequencies as explained in chapter 2.7. ‘’data analysis’’. Would you kindly provide us in which point doesn’t fulfil the mathematical sense? Of course correlations between demographic variables and individual performance of quantitative data could be more analysed (grouping them), but the length of the paper would be enormously extended. For these reasons we presented only the basic correlations.
Point 3: The implementation of Training has been insisted on for a long time, in the context of the topic of the article, which was dementia, without deciding, at least formally, the optimal age for the disease, its possible corrections, brought by medication or other therapeutic procedures.
Response 3: Thank you for this comment. We now have the chance to be more specific with the aim of the study. Prognosis, prevention, statistics on high risks groups, knowledge about pharmaceutical and non-pharmaceutical treatment, are really important issues for PwD. However, it was not a study on dementia therapy, or an intervention for PwD performance, thus conclusions or decisions about the optimal age for the disease and providing knowledge about pharmaceutical or no pharmaceutical treatment were not the aim of current study. We intended to train the staff in order to acquire better knowledge and attitude, confidence in care and therefore less anxiety in job, to help them in situations of ‘’difficult behaviours’’ during hospitalization. Beyond the age, or treatment PwD follow, staff should provide the optimal care for them with respect to their dignity. Our goal was to develop and evaluate such a training program and provide better healthcare services.
Point 4: Although a similar work is used in the bibliography, the barriers and facilitators to implementing dementia education and training in health and social care services, in Greek hospitals and what peculiarities have been observed are not highlighted at all.
Response 4: Thank you for this comment. We agree that barriers and facilitators of implementation of such programs in Greek social care services is not emphasized enough. We now added some further info supporting studies in Greek context (references: 70,71,75,79,82). You now can see the changes in chapter 4.2. lines: 129-132, 134-139 and 165-178.
Point 5: The investigation made is not of quality and does not present a novelty in the field, being a simple presentation of some data collected on various criteria.
Response 5: We would like to thank you for this last comment. We now added some additional info about the novelty of the study in ‘’conclusion’’-chapter 5, lines: 183-199.
The current study is not a pioneer in the field of dementia training implementation globally. However, it is the first training ever conducted in Greek hospital settings in which PwD are highly admitted. Till now only few trainings conducted for a minority of formal caregivers but there weren’t either evaluated/ published or even conducted in general hospitals [3–5]. Our presented data was not a simple presentation on various criteria but it followed a strict and logical process aimed to answer all research questions. Our research hypothesis reported in chapter 1.2 were: ‘’1) confidence in care, knowledge and attitudes toward dementia will be positively changed after the training 2) and these changes will be sustained over time. Moreover, 3) the state and trait of anxiety will be decreased after the training and 4) these changes will be sustained over time. Lastly it is expected 5) a positive evaluation of the whole intervention, and also 6) the implementation of what staff has learned in their daily routine’’.
Moreover our training was developed and discussed according to an evaluation and effectiveness model [6]. In this case, our data fulfils strict criteria, focused on learning, individual performance, and implementation in daily routine. Only two reviews followed a similar method to discuss their outcomes, [7,8] but the development of a training curriculum according to a specific model was not the case of any intervention study. So, the novelty is that current training was developed and evaluated according to Holton's model, a comprehensive framework which incorporates not only learning outcomes and individual performance but also the organization outcomes [6].
1/4 of hospital beds are occupied by PwD. It is an urgent need for staff to be trained, in order to provide better healthcare in those patient. Moreover, due to staff shortages and overburdened hospitals in Greece, staff is enormously stressed and dissatisfied from their job. We intended to reduce staff stress contributing in that way a novelty to the field. Moreover, setting an action plan like the recommendation of dementia training twice a year it was also a crucial initiation. More and more staff will be aware of dementia issues, minimize the stigma and learn more about how to manage difficult behaviours. In our recent ‘’educational need’’ analysis, 95% of staff working in Greek hospitals hadn’t ever trained in a dementia topic and they expressed the willingness to do it. They also expressed the expectation to enhance their clinical skills and self-efficacy in care [9]. So, the current study could be the trigger of changing culture in Greek health care services in general.
References
- Schindel Martin, L.; Gillies, L.; Coker, E.; Pizzacalla, A.; Montemuro, M.; Suva, G.; McLelland, V. An education intervention to enhance staff self-efficacy to provide dementia care in an acute care hospital in canada: A nonrandomized controlled study. Am J Alzheimers Dis Other Demen 2016, 31, 664–677, doi:10.1177/1533317516668574.
- Galvin, J.E.; Kuntemeier, B.; Al-Hammadi, N.; Germino, J.; Murphy-White, M.; McGillick, J. "Dementia-friendly hospitals: Care not crisis": An educational program designed to improve the care of the hospitalized patient with dementia. Alzheimer Dis. Assoc. Disord. 2010, 24, 372–379, doi:10.1097/WAD.0b013e3181e9f829.
- Alzheimer's Disease International. Train the trainers. https://www.alz.co.uk/train-the-trainers (accessed on 22 June 2020).
- ForAge. SET CARE:a self-study e-learning tool forformal and informal caregivers of people with dementia. http://elwg.eu/en/database/item/503-set-care-self-study-e-learning-tool-for-the-social-home-care-sector-2011-2013/.
- EUROCARERS. ECVC “Elderly Care Vocational Certificate”. https://carict.eu/servicedetail.php?id=74&qstring=dGFzaz1wcm9qZWN0cyZkYj0zJm9yZGVyPW5hbWUmZGlyPUFTQyZrZXl3b3JkPQ==,
- Holton, E.F. The flawed four-level evaluation model. Hum Resour Dev Q 1996, 7, 5–21.
- Scerri, A.; Innes, A.; Scerri, C. Dementia training programmes for staff working in general hospital settings - a systematic review of the literature. Aging Ment. Health 2016, 21, 783–796, doi:10.1080/13607863.2016.1231170.
- Gkioka, M.; Schneider, J.; Kruse, A.; Tsolaki, M.; Moraitou, D.; Teichmann, B. Evaluation and Effectiveness of Dementia Staff Training Programs in General Hospital Settings: A Narrative Synthesis with Holton’s Three-Level Model Applied. Journal of Alzheimer's Disease 2020, 78, 1089–1108, doi:10.3233/JAD-200741.
- Schneider, J.; Gkioka, M.; Papagiannopoulos, S.; Moraitou, D.; Metz, B.; Tsolaki, M.; Kruse, A.; Teichmann, B. Expectations of nursing personnel and physicians on dementia training: A descriptive survey in general hospitals in Germany and Greece. Z Gerontol Geriat 2019, 52, 249–257, doi:10.1007/s00391-019-01625-0.
Round 2
Reviewer 2 Report
The article is greatly improved in this present form.
Author Response
Dear reviewer 2,
Thank you for this comment and feedback. We think that your comments in round 1 helped us to greatly revised and improved our manuscript, in order to be easy followed by a reader.